# Sex identification in embryos and adults of Darwin's finches

**Mariya P. Dobreva**[1]*, **Joshua G. Lynton-Jenkins**[2]*, **Jaime A. Chaves**[3,4],
**Masayoshi Tokita**[5¤a], **Camille Bonneaud**[2], **Arkhat Abzhanov**[5¤b]

**1** Department of Life Sciences (Silwood Park), Imperial College London, Ascot, United Kingdom, **2** Centre for Ecology and Conservation, University of Exeter, Penryn, United Kingdom, **3** Department of Biology, San Francisco State University, San Francisco, California, United States of America, **4** Colegio de Ciencias Biológicas y Ambientales, Campus Cumbayá, Universidad San Francisco de Quito, Cumbayá, Quito, Ecuador, **5** Department of Organismic and Evolutionary Biology, Harvard University, Cambridge, Massachusetts, United States of America

¤a  Current address: Department of Biology, Toho University, Funabashi, Chiba, Japan
¤b  Current address: Department of Life Sciences (Silwood Park), Imperial College London, Ascot, United Kingdom
* mimi.pen@gmail.com (MPD); jl462@exeter.ac.uk (JGLJ)

**Data Availability Statement:** All relevant data are within the paper and its Supporting Information file.

**Funding:** MPD has received funding from the European Union's Horizon 2020 research and

## Abstract

Darwin's finches are an iconic example of adaptive radiation and evolution under natural selection. Comparative genetic studies using embryos of Darwin's finches have shed light on the possible evolutionary processes underlying the speciation of this clade. Molecular identification of the sex of embryonic samples is important for such studies, where this information often cannot be inferred otherwise. We tested a fast and simple chicken embryo protocol to extract DNA from Darwin's finch embryos. In addition, we applied minor modifications to two of the previously reported PCR primer sets for *CHD1*, a gene used for sexing adult passerine birds. The sex of all 29 tested embryos of six species of Darwin's finches was determined successfully by PCR, using both primer sets. Next to embryos, hatchlings and fledglings are also impossible to distinguish visually. This extends to juveniles of sexually dimorphic species which are yet to moult in adult-like plumage and beak colouration. Furthermore, four species of Darwin's finches are monomorphic, males and females looking alike. Therefore, sex assessment in the field can be a source of error, especially with respect to juveniles and mature monomorphic birds outside of the mating season. We caught 567 juveniles and adults belonging to six species of Darwin's finches and only 44% had unambiguous sex-specific morphology. We sexed 363 birds by PCR: individuals sexed based on marginal sex specific morphological traits; and birds which were impossible to classify in the field. PCR revealed that for birds with marginal sex specific traits, sexing in the field produced a 13% error rate. This demonstrates that PCR based sexing can improve field studies on Darwin's finches, especially when individuals with unclear sex-related morphology are involved. The protocols used here provide an easy and reliable way to sex Darwin's finches throughout ontogeny, from embryos to adults.

innovation programme under the Marie Sklodowska-Curie grant agreement No 702707. JGLJ was supported by Heredity Fieldwork Grant awarded by The Genetics Society, and by University of Exeter Vice Chancellor's scholarship for post-graduate research. JAC and CB received a field work collaboration grant by the Universidad San Francisco de Quito (Ecuador). MT was supported by Japan Society for the Promotion of Science (JSPS) – Postdoctoral Fellowship for research abroad No 23-771, and The Uehara Memorial Foundation Research Fellowship. The funders had no role in study design, data collection and analysis, decision to publish, or preparation of the manuscript.

**Competing interests:** The authors have declared that no competing interests exist.

## Introduction

Accurate and rapid sex identification is an important step in many research projects. Distinguishing sex by morphological traits like colouration and ornamentation can be straightforward in mature sexually dimorphic birds. Often however, such traits develop gradually and younger birds of both sexes look alike [1]. This makes visual sexing of nestlings, fledglings and young adults difficult to impossible. In addition, some species of birds only show clear dimorphic traits during the mating season, with males moulting into a less exaggerated plumage when not breeding [2]. Avian embryos cannot be sexed based on external observation only. Dissections for sexing based on embryonic gonadal morphology is time-consuming and might not be possible in all species, and is impossible in earlier developmental stages [3]. Furthermore, visual sexing can be difficult or impossible in monomorphic species of birds. In these cases, behavioural observations including singing, or morphometric measurements could be used, but these are not always applicable and can be expensive, inaccurate and time-consuming [4]. The presence of a brood patch can be used to recognise the incubating sex, but this is a temporary trait, not all species develop it, and it is not discriminative when both sexes incubate [5]. Cloacal protuberances in males are visible only during breeding and are not always clearly distinguishable [6].

Darwin's finches are endemic to the Galápagos and Cocos islands and represent a classical example of adaptive radiation under natural selection [7]. This unique group of birds has contributed significantly to the study of evolutionary processes. There is a constantly growing body of literature on their morphology, population genetics, genomics, behaviour, physiology, development, ecology, biomechanics, conservation, etc. [7–10]. Comparing the unfolding of genetic programs during development among different species of Darwin's finches has provided insight into the possible evolutionary mechanisms behind the extraordinary radiation of this group of birds [11–15]. Sexing embryos of Darwin's finches is important for further research, for instance when evaluating possible sex bias in comparative gene expression analyses.

Molecular sexing of post-embryonic Darwin's finches can be useful as well. It is especially relevant for birds in their first months of life, when both sexes look alike in all species [16]. In sexually dimorphic species, males develop their mature plumage colouration gradually over time—a process that usually starts around the end of their first year and takes several years [16–18]. In ground, cactus and sharp-beaked finches (genus *Geospiza*, 6 species), males develop their adult black plumage and beak colour over 4–6 years, with young males resembling the brown, streaked and pale-beaked females [8, 17]. Similarly, in tree finches (genus *Camarhynchus*, 3 species) it takes a year for the males to start developing the black colour of their heads, which is fully attained in 5 years, while females remain brown and streaked [18]. Other dimorphic species are the vegetarian finch *Platyspiza crassirostris* and the Cocos' island finch *Pinaroloxias inornata* [16]. There are four monomorphic species of Darwin's finches: the woodpecker and mangrove finches *Camarhynchus pallidus* and *C. heliobates*, and the warbler finches *Certhidea olivacea* and *C. fusca* [16]. Mature birds can be sexed by behavioural observations—especially singing—as only male Darwin's finches sing [16, 17, 19, 20]. However, males typically sing only during the mating season, and this approach is not applicable for mist-net captured birds. Only female Darwin's finches incubate the eggs, and the brood patch is a definitive female trait [16]. But brood patches and cloacal protuberances are temporary traits as well. Thus, molecular sexing is important for both juvenile Darwin's finches of all species and mature monomorphic individuals.

The heterogametic sex in birds is female with Z and W sex chromosomes, while males have two Z chromosomes [21]. Currently, the most widely used methodology for molecular sexing

in birds is polymerase chain reaction (PCR) to amplify sex chromosome-specific fragments, followed by electrophoretic analysis [4]. Earlier attempts at avian sexing involved amplification of a W-specific repetitive sequence [3, 22, 23] to allow detection of the female sex only. Later, the chromodomain helicase DNA binding 1 (*CHD1*) became the most widely used gene for non-ratite avian sex identification [4]. The Z and W chromosomes carry very closely related, but not identical copies of the *CHD1* gene [24, 25]. *CHD1Z* and *CHD1W* differ slightly in the size and sequence of some intronic regions allowing the detection of two versions of the gene in the heterogametic females and one in the homogametic males. *CHD1* is a highly conserved gene, which makes it a candidate for universal non-ratite avian sex identification. A number of studies have reported specific PCR primers to screen the intron variants of the Z and W alleles and have successfully applied these to many avian species [26–30]. The difference between the size of the male and female fragments varies between species and primer pairs. We chose two *CHD1* primer pairs reported previously: CHD1F/CHD1R [28] and P2/P8 [26]. Within passerines, the documented difference between the male and female fragments is 193–202 bp for CHD1F/CHD1R and 10–64 bp for P2/P8 [27].

PCR sexing of avian embryos was first established in chicken (*Gallus g. domesticus*)—a widely used model species in developmental biology [23]. Chicken embryos are sexed through amplification of a W-specific repetitive sequence and an 18-S ribosomal gene sequence as a PCR control, thus detecting only the female sex [3, 23, 31, 32], or via *CHD1*-specific primers [33]. Embryos of zebra finch (*Taeniopygia guttata*)—a model passerine bird species—have been sexed with *CHD1* primer pairs described by Griffiths [26], or modifications of these [34, 35]. Sexing of embryos of Darwin's finches has not been previously reported. In juveniles and adults, attempts to sex the monomorphic woodpecker finch using DNA from blood samples were unsuccessful—results from sexing based on singing did not match the results from molecular testing [19, 20]. The lack of clear sexing has led to complications of the captivity breeding plans for the endangered mangrove finch [20]. Further sexing using data from Z-linked microsatellites was used only on 10 woodpecker finches and was inconclusive for one bird (10% failure rate) [36].

Here, we isolated DNA from embryonic tissues and adult wing vein blood and modified two *CHD1* primer pairs: CHD1F/CHD1R [28], and P2/P8 [26]. We tested modified (m)P2/P8 on embryonic DNA and mCHD1F/mCHD1R on both adult and embryonic DNA of Darwin's finches. We successfully identified the sex of all tested individuals and found that sexing based on morphological characteristics can be a source of error.

## Materials and methods

### Sampling and sexing in the field

Embryos and fledged birds from natural populations were sampled under permits issued by Ministerio del Ambiente del Ecuador (Ecuadorian Ministry of the Environment, Acceso al Recurso Genético MAE-DNB-CM-2016-0041) and by Galápagos National Park (PC-08-13; PC-34-14; PC-03-18; PC-28-19). Sampling of fledged birds was also approved by University of Exeter's Research Ethics Committee (eCORN000054) and conducted according to the Animals (Scientific Procedures) Act 1986. Avian embryos of developmental stages used in this study are not regulated animals in USA and UK. Embryos (n = 29) from six species of Darwin's finches were collected on Santa Cruz and Pinta in 2013–2015 (6 *Geospiza fuliginosa*, 5 *G. fortis*, 5 *G. magnirostris*, 7 *Camarhynchus psittacula*, 1 *C. parvulus*, and 5 *Platyspiza crassirostris*) using previously described methods [37]. Only one egg per nest was collected to minimize the impact on populations. Eggs were incubated for 7 days in the field at 38°C and 60% humidity. Embryos were decapitated, immersed in RNA*later* (Sigma-Aldrich) and frozen. Fertilized

zebra finch embryos were obtained from Queen Mary University of London and incubated and stored using the same conditions. Fledged birds (n = 567) were caught using mist-nets on San Cristobal and Santa Cruz in 2018 and 2019. We sampled representatives of six species of Darwin's finches (422 *Geospiza fuliginosa*, 102 *G. fortis*, 11 *G. scandens*, 4 *Camarhynchus pallidus*, 21 *C. parvulus*, and 7 *Certhidea fusca*). Blood was collected by brachial venepuncture as described in [38]. After sampling, each bird was given water and immediately released. Where possible, we determined sex based on: plumage colouration, from female-like brown plumage to five categories of black coloration in males towards full adult black plumage [39]; the presence of a brood patch in females; and of a cloacal protuberance in males during breeding.

## DNA extraction

Embryonic DNA was extracted using the alkaline method described in [31], with subtle modifications. Firstly, a small piece of soft tissue (5–25 mg) was dissected from the head or tail regions of mid-incubation embryos preserved in RNA*later* (from stages 30–34 [40]). Samples were immersed in 40 μl 0.2N NaOH, vortexed for 15 seconds and lysed at 80˚C for 25 minutes. Samples were then placed on ice for 1 min, vortexed for 15 sec and pH-neutralized with 300 μl 0.04M Tris-HCl (pH 7.75). We used 2 μl per 20 μl PCR reaction. While most embryonic samples worked using the unpurified DNA as a template, several did not show bands in the following gel analysis. In these cases, we diluted the extracted DNA in water (1:10) and used 4 μl for PCR, which resulted in clear bands. Adult DNA was extracted from blood samples using DNeasy Blood & Tissue extraction kits (QIAGEN®) following the manufacturer's protocol for nucleated blood. Extractions were standardised to a concentration of 25ng/μl.

## PCR amplification, sequencing, and gel visualization

We aligned the sequences (BLASTn) of the CHD1F and CHD1R [28], and P2 and P8 [26] primers to those species of interest with published genomes: zebra finch (*T. guttata*) and medium ground finch (*Geospiza fortis*). We chose these primer sets because they have been assessed as most successful across passerine birds (Passeriformes) [27]. Based on the alignment results, we substituted the specified nucleotides so that the primer sequences match more closely the corresponding *CHD1* region of *T. guttata* and *G. fortis* (Table 1). We did not change the sequence of P8. The PCR reaction mixture contained 2.5 units Taq polymerase and 1 x PCR buffer (D1806, Sigma-Aldrich), 0.25 mM dNTP mix and 0.5 μM of each of the primers. For mCHD1F/mCHD1R, amplification conditions were: 4 min at 94˚C followed by 40 cycles of 94˚C for 30 sec, 56˚C for 30 sec and 72˚C for 45, and a final extension for 5 min at 72˚C. For mP2/P8, the same conditions applied but the annealing temperature used was 51˚C. PCR

**Table 1. Primer sequences.**

| Primer name elsewhere | Sequence 5'-3' |
|---|---|
| CHD1F (Lee et al. 2010) | TATCGTCAGTTTCC**T**TTTCAGGT |
| CHD1R (Lee et al. 2010) | CCTTTTATTGATCCATCAAG**C**CT |
| P8 (Griffiths et al. 1998) | CTCCCAAGGATGAGRAAYTG |
| P2 (Griffiths et al. 1998) | TCTGCATC**G**CTAAATCCTTT |
| **Primer name in this study** | **Modified sequence 5'-3'** |
| mCHD1F | TATCGTCAGTTTCC**V**TTTCAGGT |
| mCHD1R | CCTTTTATTGATCCATCAAG**T**CT |
| mP2 | TCTGCATC**R**CTAAATCCTTT |

Modifications of previously published primers are underlined.

products were visualized by electrophoresis on agarose gel containing GelRed (Biotium) and band sizes were evaluated against a 100 bp molecular weight marker (NEB). We analysed mCHD1F/mCHD1R fragments on 2% gel for 1h, 90 min and 2h at 4V/cm. mP2/P8 products were analysed on 2.5% or 3% gel for time points between 2 hours and 3 hours 30 min at 4–4.5V/cm. PCR products were sequenced by Macrogen Europe B.V. and sequences were identified using BLASTn (NCBI).

## Results

We aimed to amplify a section of the *CHD1* gene to identify the sex of nine species of Darwin's finches. Using our primer pairs (Table 1), we expected fragment size ranges (based on results from Passeriformes [27]) as follows: for mCHD1F/mCHD1R, 328–345 bp for W and 455–696 bp for Z; for mP2/P8, 339–398 bp for W and 316–371 bp for Z.

First, we analysed the DNA of embryonic samples of six species of Darwin's finches, and of the widely used passerine model species—the zebra finch (*T. guttata*). We analysed 39 embryonic samples: 10 of zebra finch and 29 embryos of Darwin's finches. DNA was extracted from embryonic tissue using the most rapid published protocol [31] with minimal modifications. All embryonic samples were amplified successfully using both mCHD1F/mCHD1R and mP2/P8 primer pairs (Table 2). Samples with a single band were identified as males, and ones with two bands—as females. Fig 1 shows the electrophoresis results of 4 zebra finch and 17 Darwin's finch embryonic samples. The mCHD1F/mCHD1R PCR products were analysed on 2% agarose gel at 4V/cm for 90 minutes for optimal band separation (Fig 1A), but shorter running times (e.g., 60 min) were also successful. To confirm the PCR results, we sequenced the PCR product of 13 of the samples, as follows: one male and one female per species for *T. guttata*, *G. magnirostris*, *G. fortis*, *G. fuliginosa*, *C. psittacula*, and *P. crassirostris*; and the only individual for *C. parvulus* (male). Table 2 shows the top BLASTn result for *CHD1* or *CHD1* homologs for each sequence, along with the E-value and the avian species with highest score. All BLASTn results for male embryos showed similarity with *CHD1Z* and not with *CHD1W*. Conversely, all results for female embryos showed similarity to both *CHD1Z* and *CHD1W*. We analysed the mP2/P8 PCR products on 3% gel at 4.25V/cm for 3 hours 30 minutes to achieve a very clear separation of the two bands, expected to have a small fragment size difference based on the data across other Passeriformes species (10–64 bp, [27]) (Fig 1B). Our tests showed that the two bands already separated on 2.5% gel at 4V/cm for 2 hours and these timesaving conditions can be used instead. Both primer pairs tested on embryonic DNA produced clearly distinguishable bands and the results obtained from the two pairs correlate to each other (Table 2).

In addition, we aimed to sex 567 mist-net captured birds of six species of Darwin's finches through a combination of morphological and genetic sexing approaches. The design of the study did not allow collection of behavioural data. Sex was clearly identified by morphology in 250 birds, based on plumage coloration and/or presence of brood patch or cloacal protuberance. 46 of these were analysed by PCR to validate the approach. All PCR sexing results matched morphological sex assignment. From the remaining 317 birds, 39 did not show any morphological traits that could be used for sexing and were not assigned sex. These included: males that had not started developing adult coloration or were monomorphic; females that were not incubating and had no brood patch; and males not in active breeding without cloacal protuberance. The other 278 birds did not show a clear sex-specific morphological trait but were assigned sex based on a partially clear, marginal trait. These could include males in early stages of development of adult colouration, or females with forming but unclear brood patch. The sex of these individuals was to be confirmed using PCR. Blood samples were amplified successfully using the mCHD1F/mCHD1R primer pair. Fig 2 shows the electrophoretic

**Table 2. List of embryonic samples analysed in Fig 1, and sexing and sequencing results.**

| Lane in Fig 1 | Sample name | P8/mP2 | mCHD1F/mCHD1R | BLAST | | | Total per species | |
|---|---|---|---|---|---|---|---|---|
| | | | | *CHD1* variant | Species | E-value | F | M |
| 1 | *Taeniopygia guttata*—1 | F | F | CHD1W | *Corvus frugilegus* | 6.E-60 | 5 | 5 |
| | | | | CHD1Z | *Taeniopygia guttata* | 1.E-42 | | |
| 2 | *Taeniopygia guttata*—2 | F | F | | | | | |
| 3 | *Taeniopygia guttata*—3 | M | M | CHD1Z | *Taeniopygia guttata* | 0.E+00 | | |
| 4 | *Taeniopygia guttata*—4 | M | M | | | | | |
| 5 | *Geospiza magnirostris*—1 | F | F | CHD1W | *Cardinalis cardinalis* | 1.E-76 | 4 | 1 |
| | | | | CHD1Z | *Motacilla flava pygmaea* | 1.E-52 | | |
| 6 | *Geospiza magnirostris*—2 | F | F | | | | | |
| 7 | *Geospiza magnirostris*—3 | M | M | CHD1Z | *Melanospiza richardsoni* | 3.E-148 | | |
| 8 | *Geospiza fortis*—1 | F | F | | | | 4 | 1 |
| 9 | *Geospiza fortis*—2 | F | F | CHD1W | *Cardinalis cardinalis* | 3.E-73 | | |
| | | | | CHD1Z | *Pomarea dimidiata* | 6.E-50 | | |
| 10 | *Geospiza fortis*—3 | M | M | CHD1Z | *Sporophila caerulescens* | 2.E-150 | | |
| 11 | *Geospiza fuliginosa*—1 | F | F | | | | 5 | 1 |
| 12 | *Geospiza fuliginosa*—2 | F | F | CHD1W | *Emberiza schoeniclus* | 1.E-27 | | |
| | | | | CHD1Z | *Oporornis tolmiei* | 3.E-48 | | |
| 13 | *Geospiza fuliginosa*—3 | M | M | CHD1Z | *Melanospiza richardsoni* | 5.E-116 | | |
| 14 | *Camarhynchus psittacula*—1 | F | F | CHD1W | *Emberiza schoeniclus* | 2.E-100 | 4 | 3 |
| | | | | CHD1Z | *Sporophila melanogaster* | 7.E-30 | | |
| 15 | *Camarhynchus psittacula*—2 | F | F | | | | | |
| 16 | *Camarhynchus psittacula*—3 | M | M | CHD1Z | *Tiaris olivacea* | 2.E-174 | | |
| 17 | *Camarhynchus psittacula*—4 | M | M | | | | | |
| 18 | *Camarhynchus parvulus*—1 | M | M | CHD1Z | *Sporophila hypoxantha* | 3.E-132 | 0 | 1 |
| 19 | *Platyspiza crassirostris*—1 | F | F | | | | 4 | 1 |
| 20 | *Platyspiza crassirostris*—2 | F | F | CHD1W | *Cardinalis cardinalis* | 8.E-69 | | |
| | | | | CHD1Z | *Oporornis philadelphia* | 1.E-52 | | |
| 21 | *Platyspiza crassirostris*—3 | M | M | CHD1Z | *Tiaris olivacea* | 1.E-175 | | |
| 22 | Negative control/Total | - | - | | | | 26 | 13 |

PCR results using two primer sets. Sequencing results are shown for one female and one male per species, as top BLASTn results. The last column represents total number of embryos analysed by PCR. F—female; M—male.

analysis on 2% agarose gel of the PCR products of 24 samples with assigned but uncertain sex. Table 3 lists sample species shown in Fig 2, assigned sex using morphology, and PCR results. Of the 24 shown individuals, one had been sexed incorrectly in the field (highlighted in bold). Table 4 represents a summary of all results from blood samples. Importantly, the PCR approach revealed a sexing error of 12.6%: from the 278 birds with marginal sex, 35 had been assigned the wrong sex in the field. The distribution of the sexing error was highly dependent on species and sample size. *C. parvulus* showed the highest error rate (50%) but very small sample size (N = 2); *G. fuliginosa* had a large sample size (N = 226) and an error rate of 8.4%; while *G. fortis* had both high error rate (30.6%) and reasonable sample size (N = 49) (Table 4).

## Discussion

In this study we applied an optimized PCR technique and successfully resolved the sex of nine species of Darwin's finches from both adult and embryonic samples. Accurate sex-

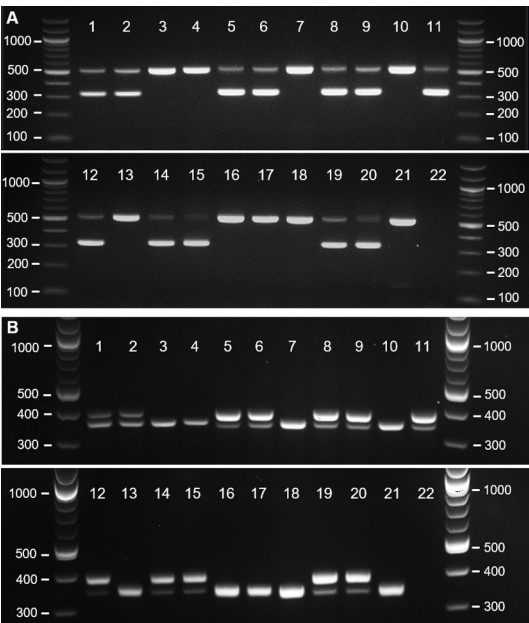

**Fig 1. Gel electrophoresis of PCR products amplified using DNA extracted from embryonic tissues.** (A) Results using the mCHD1F/mCHD1R primer pair. (B) Results using the mP2/P8 primer pair. Lane numbers correspond to the samples listed in Table 2.

determination is necessary for a range of biological applications. For example, comparative transcriptomics, such as RNA-seq, is a vital tool in developmental biology. Sex-specific variation in gene expression can introduce bias when individual embryo specimens are compared. In multi-species comparisons, such as those on Darwin's finches, sex-related differences might be misinterpreted as inter-specific variation. Accounting for specimen sex is therefore essential for the accurate interpretation of expression data. In mice, sexually dimorphic gene expression in embryos starts as soon as the embryonic genome is activated at the two cell stage [41]. In chickens, such dimorphic expression is documented from at least as early as the blastoderm stage [42]. However, unlike the mouse model, where the expression profiles of many X- and Y-linked genes are known [41], fewer W-specific genes are characterised in birds and their expression levels are tissue-specific and not known for many tissue types and stages [42]. Therefore, it can be challenging to determine sex in birds from RNA-seq data alone. PCR sexing from DNA might prove easier, faster and more reliable [43] especially in non-model organisms. Comparative studies based on fixed tissues are widely used in developmental biology, such as *in situ* hybridization and immunohistochemical stainings. While the DNA

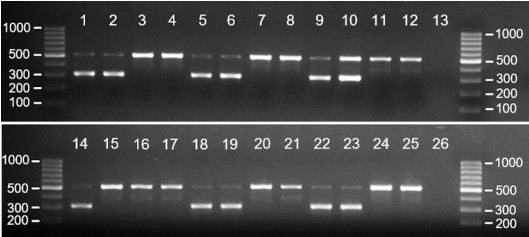

**Fig 2. Gel electrophoresis of PCR products amplified using mCHD1F/mCHD1R primer pair and DNA extracted from blood.** Lane numbers correspond to the samples listed in Table 3.

**Table 3. Samples showed in Fig 2: Comparison between morphological and molecular sex identification using the mCHD1F/mCHD1R primer pair.**

| Lane in Fig 2 | Species | Sex based on morphology | Sex based on PCR |
|---|---|---|---|
| 1 | *Geospiza fortis—1* | F | F |
| 2 | *Geospiza fortis—2* | F | F |
| 3 | *Geospiza fortis—3* | M | M |
| 4 | *Geospiza fortis—4* | M | M |
| 5 | *Geospiza fuliginosa—1* | F | F |
| 6 | *Geospiza fuliginosa—2* | F | F |
| 7 | *Geospiza fuliginosa—3* | M | M |
| 8 | *Geospiza fuliginosa—4* | M | M |
| 9 | *Geospiza scandens—1* | F | F |
| 10 | *Geospiza scandens—2* | F | F |
| 11 | *Geospiza scandens—3* | M | M |
| 12 | *Geospiza scandens—4* | M | M |
| 13 | Negative control | - | - |
| 14 | *Camarhynchus pallidus—1* | F | F |
| **15** | ***Camarhynchus pallidus—2*** | **F** | **M** |
| 16 | *Camarhynchus pallidus—3* | M | M |
| 17 | *Camarhynchus pallidus—4* | M | M |
| 18 | *Camarhynchus parvulus—1* | F | F |
| 19 | *Camarhynchus parvulus—2* | F | F |
| 20 | *Camarhynchus parvulus—3* | M | M |
| 21 | *Camarhynchus parvulus—4* | M | M |
| 22 | *Certhidea fusca—1* | F | F |
| 23 | *Certhidea fusca—2* | F | F |
| 24 | *Certhidea fusca—3* | M | M |
| 25 | *Certhidea fusca—4* | M | M |
| 26 | Negative control | - | - |

F—female; M—male.

**Table 4. Adult and juvenile Darwin's finches captured and sexed by morphology or later resolved by PCR.**

| Species | Total captured | Sexing by morphology | | | | | | | | | Sexing by PCR | | | | | | |
|---|---|---|---|---|---|---|---|---|---|---|---|---|---|---|---|---|---|
| | | Clear trait | | | Marginal trait | | | No trait | All applications | | | Misassigned sex | | | Sexing error |
| | | *Total* | F | M | *Total* | F | M | *Total* | *Total* | F | M | *Total* | F | M | % |
| *Geospiza fuliginosa* | 422 | *175* | 44 | 131 | *226* | 179 | 47 | *21* | *271* | 191 | 80 | *19* | 4 | 15 | 8.4 |
| *Geospiza fortis* | 102 | *51* | 17 | 34 | *49* | 40 | 9 | *2* | *57* | 27 | 30 | *15* | 0 | 15 | 30.6 |
| *Certhidea fusca*\* | 7 | *6* | 2 | 4 | *0* | 0 | 0 | *1* | *6* | 3 | 3 | *0* | 0 | 0 | - |
| *Camarhynchus pallidus*\* | 4 | *1* | 0 | 1 | *0* | 0 | 0 | *3* | *4* | 1 | 3 | *0* | 0 | 0 | - |
| *Camarhynchus parvulus* | 21 | *7* | 1 | 6 | *2* | 1 | 1 | *12* | *16* | 6 | 10 | *1* | 0 | 1 | 50 |
| *Geospiza scandens* | 11 | *10* | 5 | 5 | *1* | 0 | 1 | *0* | *9* | 4 | 5 | *0* | 0 | 0 | 0 |
| Total | 567 | *250* | 69 | 181 | *278* | 220 | 58 | *39* | *363* | 232 | 131 | *35* | 4 | 31 | 12.6 |

Birds with marginal sex-specific morphology or undetermined sex were analysed by PCR. The sexing error represents the percentage of birds that were mis-assigned based on marginal sex-specific morphology. Numbers for mis-assigned sex reflect the PCR-determined sex, e.g., a count for male indicates a bird incorrectly sexed as a female in the field. PCR across all applications refers to birds with marginal or no sex-specific traits and birds with clear sex morphology sexed as proof of principle controls.

\*Monomorphic species.

extracted from fixed tissues is often low in quantity and integrity, and therefore unsuitable for most applications, PCR amplification of relatively short DNA fragments remains viable [44]. Sexing by gonadal differentiation is possible in chicken embryos after a certain embryonic stage [3]. However, it requires dissection and is a laborious procedure to carry out for large numbers of individuals, particularly for embryos of smaller avian species. By comparison, molecular sexing is a much cheaper and faster alternative.

Ideally, DNA extraction and PCR reactions for sex genotyping, should be rapid and simple. Protocols to extract DNA from embryonic tissue that do not involve long protein digestion and cleaning steps are preferred and work equally well as lengthy overnight procedures [31]. Here, we apply the so-called alkaline method used previously on chicken embryos [31], with minor modifications. Alkaline lysis relies on solubilisation of proteins while DNA remains stable [45] and is a fast and simple strategy to obtain DNA from small amounts of tissues—it takes only 30 minutes. For fledged birds, we used the rapid and effective standard procedures of blood collection by brachial venepuncture [38] followed by DNA extraction. Where less invasive methods are required, e.g. for younger birds such as nestlings, DNA extracted from buccal swabs can be used for sexing [46].

Different primer sets based on the *CHD1* gene have been used to sex birds [26, 28–30]. We chose two that have been successful in many passerines [27] and introduced minor modifications to their sequences. Our results confirm that both primer sets work clearly and easily to identify the sex of Darwin's finches (Figs 1 and 2).

There are two advantages to the molecular sexing of post-embryonic Darwin's finches. Firstly, juveniles, such as nestlings, fledglings, and young adults—usually in their first year of life, look alike. Secondly, in monomorphic species of Darwin's finches the colour of plumage and beak are the same in both sexes. Non-breeding birds lack traits such as protruding cloaca in males and brood patch in females, or they can be unclear. In these cases, molecular sexing can be used for either sex identification, or confirmation.

Strictly speaking, all Darwin's finches are sexually dimorphic in terms of size. Across the clade, male body and bill sizes are on average slightly larger than those of females [17, 47]. Interestingly, female warbler finches (*Certhidea olivacea* and *C. fusca*) have longer beaks than the males, which is a peculiar case of reversed sexual dimorphism where the directions of beak and body size dimorphisms do not match [47]. However, even though they are significant at the population level, differences in size cannot be used for accurate sexing because of the large marginal area where male and female individual sizes can overlap.

Molecular sexing might not be needed in long-term studies where individual birds are being followed through their lives and there is enough morphological and behavioural data [8]. This is especially true for dimorphic species, but even monomorphic species could be sexed fairly confidently based on mating behaviour, e.g. males building nests and singing, females assessing the nests and male fitness [8]. However, many studies, including this one, involve single capture and release in which case behavioural data and temporary traits may not be available. Attempts at molecular sexing of woodpecker and mangrove finches have been unsatisfactory, causing difficulties with the mangrove finch recovery plans [19, 20, 36]. The approach described here has the potential to save time and resources in future conservation projects. The only report on PCR sexing of Darwin's finches we are aware of uses primers described by Griffiths et al. [26] and includes 68 juvenile tree finches [48]. We successfully sexed 363 post-embryonic birds from six species of Darwin's finches, both dimorphic and monomorphic, and provide detailed optimised protocols.

As demonstrated by our study, visual sexing of Darwin's finches can be a source of uncertainty and can introduce significant error. This happens especially often with juveniles that have not yet developed sufficiently clear sexual dimorphism, and with birds that are studied

outside of the mating season. It is of note that only 44% of all caught birds were sexed with confidence in the field based on morphological traits. In our hands, PCR sexing of the 42% "marginal" individuals revealed 13% rate of wrong sex assignment in the field across all species (Table 4). The error rate varied considerably between species, but so did the sample size. It is clear however that for *G. fortis*, where the sample size is adequate, the error rate of 31% is considerable. Males were misidentified much more frequently than females, which could be expected, as immature males resemble females. In addition, 7% of all caught birds were impossible to sex at all by morphology. Lastly, the experience of the handlers in the field could affect error rate. Less experience might result in greater error, but experienced handlers are not always available. In our experience, some error will persist even after years of handling. Our results show that post-embryonic PCR sexing is useful to both confirm uncertain sexing observations, and to identify sex. It enables cheap and rapid sexing whenever sex cannot be inferred from existing data (e.g., sequenced genomes) thus improving field studies on Darwin's finches.

In conclusion, we describe sexing of multiple individuals and species of Darwin's finches based on optimised existing protocols, easily and reliably and throughout ontogeny.

## Supporting information

**S1 Raw images.**
(TIFF)

## Acknowledgments

We want to thank Julia George and David Clayton at Queen Mary University of London who provided us with fertilized zebra finch eggs. We thank: Charles Darwin Research Station, Galápagos, for assistance with permit applications, administrative and logistical support; the Galápagos Science Center for support during laboratory work; and Galápagos Institute of Arts and Sciences (GAIAS)–Universidad San Francisco de Quito for logistical support.

## Author Contributions

**Conceptualization:** Mariya P. Dobreva.

**Formal analysis:** Mariya P. Dobreva, Joshua G. Lynton-Jenkins.

**Investigation:** Mariya P. Dobreva, Joshua G. Lynton-Jenkins.

**Methodology:** Mariya P. Dobreva.

**Project administration:** Mariya P. Dobreva.

**Resources:** Mariya P. Dobreva, Joshua G. Lynton-Jenkins, Jaime A. Chaves, Masayoshi Tokita, Camille Bonneaud, Arkhat Abzhanov.

**Supervision:** Jaime A. Chaves, Camille Bonneaud, Arkhat Abzhanov.

**Validation:** Joshua G. Lynton-Jenkins.

**Visualization:** Mariya P. Dobreva.

**Writing – original draft:** Mariya P. Dobreva.

**Writing – review & editing:** Mariya P. Dobreva, Joshua G. Lynton-Jenkins, Jaime A. Chaves, Masayoshi Tokita, Camille Bonneaud, Arkhat Abzhanov.

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
