## [Decision Letter · Decision Letter 0]

26 Oct 2020

PONE-D-20-22782

Sex identification in embryos and adults of Darwin’s finches

PLOS ONE

Dear Dr. Dobreva,

Thank you for submitting your manuscript to PLOS ONE. Your study has now been evaluated by three reviewers. As you will see all reviewers find your work of interest but also raise a number of points that would need to be addressed before publication can be considered. Therefore, I return your manuscript to you for making a revision. Please, include with your revised version a detailed response on how your revision addresses the points raised during the review process.

We look forward to receiving your revised manuscript.

Kind regards,

Anton Wutz

Academic Editor

PLOS ONE

Journal Requirements:

2.PLOS ONE now requires that authors provide the original uncropped and unadjusted images underlying all blot or gel results reported in a submission’s figures or Supporting Information files. This policy and the journal’s other requirements for blot/gel reporting and figure preparation are described in detail at https://journals.plos.org/plosone/s/figures#loc-blot-and-gel-reporting-requirements and https://journals.plos.org/plosone/s/figures#loc-preparing-figures-from-image-files. When you submit your revised manuscript, please ensure that your figures adhere fully to these guidelines and provide the original underlying images for all blot or gel data reported in your submission. See the following link for instructions on providing the original image data: https://journals.plos.org/plosone/s/figures#loc-original-images-for-blots-and-gels.

Reviewers' comments:

Reviewer's Responses to Questions

**Comments to the Author**

1. Is the manuscript technically sound, and do the data support the conclusions?

Reviewer #1: Partly

Reviewer #2: Yes

Reviewer #3: Yes

2. Has the statistical analysis been performed appropriately and rigorously? 

Reviewer #1: N/A

Reviewer #2: N/A

Reviewer #3: Yes

3. Have the authors made all data underlying the findings in their manuscript fully available?

Reviewer #1: Yes

Reviewer #2: Yes

Reviewer #3: Yes

4. Is the manuscript presented in an intelligible fashion and written in standard English?

Reviewer #1: Yes

Reviewer #2: Yes

Reviewer #3: Yes

5. Review Comments to the Author

Reviewer #1: The sexing of samples is a crucial step before performing, for example, differential expression analyses. Sexing embryos is achieved using gonadal morphology or the amplification of specific sequences from sex chromosomes. Visual sexing of adult birds might be complex too, particularly if males and females are not dimorphic. In this study, the authors used two pairs of primers that have been widely used to sex passerines to verify that they can also be used to correctly sex six species of Darwin’s finches (embryos and adults). The study is well-made and could be important to perform sex-specific studies in Darwin’s finches.

Specific comments:

Page 7, line 94: Please clarify in the Methods the number of embryos used for each of the six species.

Page 7, line 96: Please use Celsius instead of Fahrenheit.

Page 7, line 99: Please clarify in the Methods how many adults were captured for each of the six species. Also, please clarify whether the individuals were healed and immediately released after the blood samples were taken.

Page 9, line 131: Please add the experiments, methods, and references you used to define that the nucleotide changes you highlighted on the primer’s sequences (Table 1) would “make them more specific to Darwin’s finches”.

Figure 1: Given the limited number of embryos analyzed (29), why not showing the results of the PCR amplification for all of them? Currently PCR data for 17 embryos is displayed in Figure 1.

General comments:

Since the main objective of this work was to show that a modified set of primers could correctly sex embryos of Darwin’s finches, it would have been ideal to also show that the PCR banding would perfectly match gonadal morphology of the 29 embryos. PCR amplification in Zebra finch was used as a positive control, but, again, the sex of these embryos was not validated using an alternative technique. Strictly speaking, using the PCR results from other bird species as control may be insufficient since CDH1Z and CDH1W in Darwin’s finches may have followed gene conversion, gene duplication, gene deletion, etc. I understand that primers designed to amplify CDH1Z and CDH1W have been widely used to sex numerous bird species and sexing of Darwin’s finches in the study is likely accurate. However, it seems that validating the PCR results using gonadal morphology would have been the perfect control for this project.

Alternatively, if gonadal morphology is no longer available, the authors could perform Sanger sequencing of a few PCR products and show that male amplicons correspond unequivocally to CDH1Z whereas female amplicons correspond to CDH1Z and CDH1W.

Accurate embryo sexing is particularly important since the authors later explain that the error rate of sexing adults based on morphological traits is not negligible and, therefore, using morphological traits is also not a good control.

Reviewer #2: PONE-D-20-2278

Sex identification in embryos and adults of Darwin’s finches.

Dobreva and colleagues carried out a study involving molecular sexing in a range of Darwin’s finches species. The study brings an optimized set of primers, from previously described primers often used for molecular sexing in other bird groups. Jointly with modifications in some steps of DNAs extraction, the fast protocol proved to be an useful tool in accessing the sex information from hatchlings to adults of a group of birds that in most of the cases look alike.

The study is worth to be published. I have only minor comments detailed below.

Lines 94-97

Embryos from six species of Darwin’s finches were collected on Santa Cruz and Pinta in 2013-2015 (Geospiza fuliginosa, G. fortis, G. magnirostris, Camarhynchus psittacula, C. parvulus, and Platyspiza crassirostris) using previously described methods [37]. Briefly, the third egg of each nest was incubated for 7 days in the field at 100°F and 60% humidity. The embryo was then decapitated, immersed in RNAlater (Sigma-Aldrich) and frozen.

Why did the authors incubate solely the third egg of each nest? Are the results different comparing with the embryos? Or is there some specific reason for this procedure? If yes, could the authors clarify it in the text?

Lines 104-105

Embryos and fledged birds were sampled from natural populations under permits and according to regulations established by the Ecuadorian Ministry of the Environment and Galapagos National Park. Birds were captured and sampled under University of Exeter's Research Ethics Committee approval (eCORN000054).

I suggest to change the sentence: “Embryos and fledged birds were sampled from natural populations” to “Embryos and fledged birds from natural populations were sampled”, once this may lead the readers to believe that all animals were sampled in field activities, but in the previous sentence the authors say that some samples were also obtained from the University of Londres, in the lines 97-98.

Line 107

Embryonic DNA was extracted, with modifications, using the “alkaline method” described elsewhere [31].

I suggest to reword this sentence as: Embryonic DNA was extracted using “alkaline method” following the details described in [31], with subtle modifications. Firstly, a small piece of soft tissue… …

Lines 204 - 205

Here, we applied the so-called alkaline method used previously on chicken embryos,

with minor modifications [31].

I suggest to rephrase this sentence as: “Here, we applied the so-called alkaline method used previously on chicken embryos, [31] with minor modifications.” Since the modifications are not from the [31] study.

Reviewer #3: The goal of this study is to validate the use of a modified set of PCR primers for molecular sexing of several Darwin’s finch species, including both embryonic and juvenile/adult samples. The authors have demonstrated that these new primers accurately identify sex, and may be a critical tool for various aspects of study, including conservation work. However, the justification for the development of new PCR primers is not clear. It was not clear if the widely used set of primers for CHD1, from which the primers used here are derived, failed to work in Darwin’s finches, as the citations used to support this simply say that genetic sexing does not work without any further clarification. Furthermore, another citation used the previous set of primers to sex juvenile tree finches. Finally, there needs to be more detail regarding how the primers were modified, including where the new sequence data come from and why the modifications to the primers were made. To conclude, while I think this is a technically sound paper, it failed to convince me that a new set of primers for molecular sexing was necessary in the first place.

Line 27-28: “For birds with marginal sex specific traits, PCR results revealed a 13% sexing error rate.” Be more clear here that birds in the field were incorrectly sexed. In my initial reading I thought this meant the PCR didn’t work properly for 13% of individuals.

Line 55: Should be 4-6 years?

Line 81: Inappropriate “—”

Line 83-86: Be more specific about why previous attempts at molecular sexing have not worked. Did previous studies use the same markers used in this study?

Line 99: 567 adult individuals or embryos and adults?

Line 111: “When no bands were visible, 4 μl of a 1:10 dilution in water were used for PCR” Unclear what this means.

Line 115-117: Where did the CHD1 data for T. guttata and G. fortis come from? Is this previously published data or newly published data? What modifications were made to the primers? This is the essence of the paper so there needs to be a little more clarity here for how new primers were designed. Did the P8 primer not need any modification?

Line 158: missing an “and”.

Line 186: The beginning of the discussion begins with comparative transcriptomics, but this was never discussed elsewhere in the paper. Perhaps better to start off with a more general need for confident sex identification, particularly for developing embryos or species without sexually dimorphic traits.

Line 228-229: Be more clear about how the methods used here improve on the failed methods of molecular sexing of the mangrove finch.

Line 231: Citation format is odd regarding Griffiths et al. and then a different citation.

6. PLOS authors have the option to publish the peer review history of their article (what does this mean?). If published, this will include your full peer review and any attached files.

Reviewer #1: No

Reviewer #2: No

Reviewer #3: No

---

## [Author Response · Author response to Decision Letter 0]

19 Jan 2021

We thank all reviewers for their detailed comments and suggestions. All page/line references match the manuscript version with track changes.

Reviewer #1

The sexing of samples is a crucial step before performing, for example, differential expression analyses. Sexing embryos is achieved using gonadal morphology or the amplification of specific sequences from sex chromosomes. Visual sexing of adult birds might be complex too, particularly if males and females are not dimorphic. In this study, the authors used two pairs of primers that have been widely used to sex passerines to verify that they can also be used to correctly sex six species of Darwin’s finches (embryos and adults). The study is well-made and could be important to perform sex-specific studies in Darwin’s finches.

• We thank the reviewer for their positive evaluation of our work.

Specific comments

Page 7, line 94: Please clarify in the Methods the number of embryos used for each of the six species.

• We added the numbers. The sentence now reads: “Embryos (n=29) from six species of Darwin’s finches were collected on Santa Cruz and Pinta in 2013-2015 (6 Geospiza fuliginosa, 5 G. fortis, 5 G. magnirostris, 7 Camarhynchus psittacula, 1 C. parvulus, and 5 Platyspiza crassirostris) using previously described methods [37].” (Page 7, line 102-104)

Page 7, line 96: Please use Celsius instead of Fahrenheit.

• This sentence now reads: “Briefly, the third egg of each nest was incubated for 7 days in the field at 38°C and 60% humidity.” (Page 7, line 105)

Page 7, line 99: Please clarify in the Methods how many adults were captured for each of the six species. Also, please clarify whether the individuals were healed and immediately released after the blood samples were taken.

• The number of captured birds per species is given in Table 4 but following the reviewer’s suggestion we added them to the Methods section too to improve readability: “We sampled representatives of six species of Darwin’s finches (422 Geospiza fuliginosa, 102 G. fortis, 11 G. scandens, 4 Camarhynchus pallidus, 21 C. parvulus, and 7 Certhidea fusca).” We added a sentence on the treatment of the birds after sampling: “After sampling, each bird was given water and then immediately released.” (page 8, line 111)

Page 9, line 131: Please add the experiments, methods, and references you used to define that the nucleotide changes you highlighted on the primer’s sequences (Table 1) would “make them more specific to Darwin’s finches”.

• We re-formulated this paragraph. We aligned the sequences of the published primers with the T. guttata and G. fortis genomes in BLASTn. As these primers are used for a wide variety of birds, we tested how similar are their sequences to our species of interest with published genomes (T. guttata and G. fortis). Based on the alignment results, we chose to substitute the specified nucleotides as we speculated this would improve specificity. We have not tested the original primers and cannot provide comparisons between original and modified primer pairs. To avoid repetitiveness, we now discuss this part only in Materials and Methods. The paragraph now reads: “We aligned the sequences (BLASTn) of the CHD1F and CHD1R [28], and P2 and P8 [26] primers to those species of interest with published genomes: zebra finch (T. guttata) and medium ground finch (Geospiza fortis). We chose these primer sets because they have been assessed as most successful across passerine birds (Passeriformes)[27]. Based on the alignment results, we substituted the specified nucleotides so that the primer sequences match more closely the corresponding CHD1 region of T. guttata and G.fortis (Table 1). We did not change the sequence of P8.” (Page 9, Line 128-133).

Figure 1: Given the limited number of embryos analyzed (29), why not showing the results of the PCR amplification for all of them? Currently PCR data for 17 embryos is displayed in Figure 1.

• We aimed to display a similar number of embryonic and adult samples in order to achieve a clear, detailed enough, and yet concise representation of the PCR results. Similar publications often include only one specimen per sex per species. We aimed to include two individuals per sex per species whenever possible (and less for some species, depending on availability). If the reviewer wishes, we could provide the full results as a separate attachment for their attention.

General comments

Since the main objective of this work was to show that a modified set of primers could correctly sex embryos of Darwin’s finches, it would have been ideal to also show that the PCR banding would perfectly match gonadal morphology of the 29 embryos. PCR amplification in Zebra finch was used as a positive control, but, again, the sex of these embryos was not validated using an alternative technique. Strictly speaking, using the PCR results from other bird species as control may be insufficient since CDH1Z and CDH1W in Darwin’s finches may have followed gene conversion, gene duplication, gene deletion, etc. I understand that primers designed to amplify CDH1Z and CDH1W have been widely used to sex numerous bird species and sexing of Darwin’s finches in the study is likely accurate. However, it seems that validating the PCR results using gonadal morphology would have been the perfect control for this project.

Alternatively, if gonadal morphology is no longer available, the authors could perform Sanger sequencing of a few PCR products and show that male amplicons correspond unequivocally to CDH1Z whereas female amplicons correspond to CDH1Z and CDH1W.

Accurate embryo sexing is particularly important since the authors later explain that the error rate of sexing adults based on morphological traits is not negligible and, therefore, using morphological traits is also not a good control.

We thank the reviewer for this thoughtful suggestion. Gonadal morphology would be a great control, but it is not applicable in our case. The 10 zebra finch embryos we used were too developmentally young to distinguish male from female gonads. In both zebra finch and chicken, gonadal morphology can be reliably used for sexing from stage 32-33 onwards (7.5-8 days of incubation) (Jung et al., 2019, FASEB 33: 13825-13836. doi:10.1096/fj.201900760RR; Clinton et al., 2001, British Poultry Science, 42:1, 134-138, DOI: 10.1080/713655025). Our embryos ranged from stages 25 to 32 (one embryo at 32). For Darwin’s finches, we had only collected the heads as per our Galapagos permits. To comply with the reviewer’s comment, we performed Sanger sequencing of 13 PCR products represented in Fig. 1 (one per sex per species) and updated Table 2 to include the sequencing results. We included the following part in Results: “To confirm the PCR results, we sequenced the PCR product of 13 of the samples, as follows: one male and one female per species for T. guttata, G. magnirostris, G. fortis, G. fuliginosa, C. psittacula, and P. crassirostris; and the only individual for C. parvulus (male). Table 2 shows the top BLASTn result for CHD1 or CHD1 homologs for each sequence, along with the E-value and the avian species with highest score. All BLASTn results for male embryos showed similarity with CHD1Z and not with CHD1W. Conversely, all results for female embryos showed similarity to both CHD1Z and CHD1W.” (Page 11, Lines 160-164).

Reviewer #2

Dobreva and colleagues carried out a study involving molecular sexing in a range of Darwin’s finches species. The study brings an optimized set of primers, from previously described primers often used for molecular sexing in other bird groups. Jointly with modifications in some steps of DNAs extraction, the fast protocol proved to be an useful tool in accessing the sex information from hatchlings to adults of a group of birds that in most of the cases look alike.

The study is worth to be published. I have only minor comments detailed below.

• We thank the reviewer for their positive evaluation of our work.

Lines 94-97: Embryos from six species of Darwin’s finches were collected on Santa Cruz and Pinta in 2013-2015 (Geospiza fuliginosa, G. fortis, G. magnirostris, Camarhynchus psittacula, C. parvulus, and Platyspiza crassirostris) using previously described methods [37]. Briefly, the third egg of each nest was incubated for 7 days in the field at 100°F and 60% humidity. The embryo was then decapitated, immersed in RNAlater (Sigma-Aldrich) and frozen. Why did the authors incubate solely the third egg of each nest? Are the results different comparing with the embryos? Or is there some specific reason for this procedure? If yes, could the authors clarify it in the text?

• There is a specific reason to collect only the third egg per nest. Darwin’s finches are endemic to Galapagos and are under strict control by the Galapagos National Park. We are allowed to collect eggs if this does not affect the breeding pairs. Finches lay one egg per day and if the first one or two are taken, parents may abandon the nest, as it is a sign that predators have found it. If the 3rd or 4th egg is taken, the female lays a replacement egg to complete the clutch (clutch size is usually 3-5). Thus, this strategy may have zero consequences for breeding success. As we found this explanation too long for the purposes of the paper, we edited the sentence as follows: “Only one egg per nest was collected to minimize the impact on populations.” (Line 104)

Lines 104-105: Embryos and fledged birds were sampled from natural populations under permits and according to regulations established by the Ecuadorian Ministry of the Environment and Galapagos National Park. Birds were captured and sampled under University of Exeter's Research Ethics Committee approval (eCORN000054). I suggest to change the sentence: “Embryos and fledged birds were sampled from natural populations” to “Embryos and fledged birds from natural populations were sampled”. Once this may lead the readers to believe that all animals were sampled in field activities, but in the previous sentence the authors say that some samples were also obtained from the University of Londres, in the lines 97-98.

• We edited the sentence according to the reviewer’s suggestion and now it reads: “Embryos and fledged birds from natural populations were sampled under permits...” (Line 98)

Line 107: Embryonic DNA was extracted, with modifications, using the “alkaline method” described elsewhere [31]. I suggest to reword this sentence as: Embryonic DNA was extracted using “alkaline method” following the details described in [31], with subtle modifications. Firstly, a small piece of soft tissue…

• We edited the sentence as follows: “Embryonic DNA was extracted using the alkaline method described in [31], with subtle modifications. Firstly, a small piece of soft tissue (5-25 mg) was dissected from the head or tail regions of mid-incubation embryos preserved in RNAlater (from stages 30-34 [40]).” (Lines 118-120)

Lines 204-205: Here, we applied the so-called alkaline method used previously on chicken embryos,

with minor modifications [31]. I suggest to rephrase this sentence as: “Here, we apply the so-called alkaline method used previously on chicken embryos, [31] with minor modifications.” Since the modifications are not from the [31] study.

• We edited the sentence following the reviewer’s suggestion: “Here, we apply the so-called alkaline method used previously on chicken embryos [31], with minor modifications.” (Lines 235-236).

Reviewer #3

The goal of this study is to validate the use of a modified set of PCR primers for molecular sexing of several Darwin’s finch species, including both embryonic and juvenile/adult samples. The authors have demonstrated that these new primers accurately identify sex, and may be a critical tool for various aspects of study, including conservation work. However, the justification for the development of new PCR primers is not clear. It was not clear if the widely used set of primers for CHD1, from which the primers used here are derived, failed to work in Darwin’s finches, as the citations used to support this simply say that genetic sexing does not work without any further clarification. Furthermore, another citation used the previous set of primers to sex juvenile tree finches. Finally, there needs to be more detail regarding how the primers were modified, including where the new sequence data come from and why the modifications to the primers were made. To conclude, while I think this is a technically sound paper, it failed to convince me that a new set of primers for molecular sexing was necessary in the first place.

• We thank the reviewer for their useful input. We addressed the issues that they pointed out bellow.

Line 27-28: “For birds with marginal sex specific traits, PCR results revealed a 13% sexing error rate.” Be more clear here that birds in the field were incorrectly sexed. In my initial reading I thought this meant the PCR didn’t work properly for 13% of individuals.

• We thank the reviewer for this important remark! We rephrased the sentence as follows: “PCR revealed that for birds with marginal sex specific traits, sexing in the field produced a 13% error rate.” (Line 28-29)

Line 55: Should be 4-6 years?

• We edited the sentence as follows: “In ground, cactus and sharp-beaked finches (genus Geospiza, 6 species), males develop their adult black plumage and beak colour over 4-6 years, with young males resembling the brown, streaked and pale-beaked females [8,17].” (Line 57)

Line 81: Inappropriate “—”

• We replaced with the correct use of m-dash (Line 83).

Line 83-86: Be more specific about why previous attempts at molecular sexing have not worked. Did previous studies use the same markers used in this study?

• It is not clear from the cited studies why exactly sexing has not worked. We extracted all available information on the subject from the publications, but it was very limited. The studies have not used our primers, as they were modified by us for this study. References [19] and [20] are Husbandry guidelines for the woodpecker finch and Recovery plan for the mangrove finch. According to both, results from sexing based on singing (only males sing) did not match molecular sexing using DNA from blood samples. However, no protocols are mentioned and it is unclear what method the authors have used. To the best of our knowledge, there are no later publications by the “mangrove team”, that include sexing methodology and results. For reference [36], we included additional information and the sentence now reads: “Further sexing using data from Z-linked microsatellites was used only on 10 woodpecker finches and was inconclusive for one bird (10% failure rate) [36].” (Line 88-89)

Line 99: 567 adult individuals or embryos and adults?

• 567 adults (we use “fledged” or “post-embryonic” birds as these include both adults and juveniles) and 29 embryos. We rephrased this part to make it clearer: “Embryos (n=29) from six species of Darwin’s finches were collected…” (Line 102), and “Fledged birds (n=567) were caught using mist-nets on San Cristobal and Santa Cruz in 2018 and 2019. We sampled representatives of six species of Darwin’s finches…” (Lines 108-110)

Line 111: “When no bands were visible, 4 μl of a 1:10 dilution in water were used for PCR” Unclear what this means.

• We worked with unknown DNA concentrations as we used unpurified DNA directly for PCR. We did not see bands for several embryonic samples. Based on our experience we speculated this might be due to a high concentration of DNA and/or enzyme inhibitors in the lysate. When DNA was diluted in water (1 volume DNA in 10 volumes water), all samples showed clear bands. We realised the sentence was not clear and now it reads: “While most embryonic samples worked using the unpurified DNA as a template, several did not show bands in the following gel analysis. In these cases, we diluted the extracted DNA in water (1:10) and used 4 μl for PCR, which resulted in clear bands.” (Lines 122-124)

Line 115-117: Where did the CHD1 data for T. guttata and G. fortis come from? Is this previously published data or newly published data? What modifications were made to the primers? This is the essence of the paper so there needs to be a little more clarity here for how new primers were designed. Did the P8 primer not need any modification?

• The data for the CHD1 for T. guttata and G. fortis comes from their published genomes (NCBI). We aligned the sequences of the published CHD1R/F and P2/P8 primers with these genomes in BLASTn. As these primers are used for a wide variety of birds, we tested how similar are their sequences to T. guttata and G. fortis (one of the Darwin’s finches), as these are our species of interest that have published genomes. Based on the alignment results, we modified the primers and the modifications are shown in Table 1. We substituted the specified nucleotides as we speculated this would improve specificity. We have not tested the original primers and cannot provide comparisons between original and modified primer pairs. To avoid repetitiveness, we now discuss this part only in Materials and Methods. The paragraph now reads: “We aligned the sequences (BLASTn) of the CHD1F and CHD1R [28], and P2 and P8 [26] primers to those species of interest with published genomes: zebra finch (T. guttata) and medium ground finch (Geospiza fortis). We chose these primer sets because they have been assessed as most successful across passerine birds (Passeriformes)[27]. Based on the alignment results, we substituted the specified nucleotides so that the primer sequences match more closely the corresponding CHD1 region of T. guttata and G.fortis (Table 1). We did not change the sequence of P8.” (Page 9, Line 128-133).

Line 158: missing an “and”.

• “And” was added: “…females that were not incubating and had no brood patch; and males not in active breeding without cloacal protuberance.” (Line 184)

Line 186: The beginning of the discussion begins with comparative transcriptomics, but this was never discussed elsewhere in the paper. Perhaps better to start off with a more general need for confident sex identification, particularly for developing embryos or species without sexually dimorphic traits.

• We agree that the introduction to Discussion needed to be improved. It now reads: “In this study we applied an optimized PCR technique and successfully resolved the sex of nine species of Darwin’s finches from both adult and embryonic samples. Accurate sex-determination is necessary for a range of biological applications. For example, comparative transcriptomics, such as RNA-seq, is a vital tool in developmental biology.” (Line 212-2014).

Line 228-229: Be more clear about how the methods used here improve on the failed methods of molecular sexing of the mangrove finch.

• We do not have details on the methods used for the woodpecker and mangrove finch, besides the ones mentioned in the text. We hope our approach will be used for future mangrove finch conservation efforts. (Lines 257-260)

Line 231: Citation format is odd regarding Griffiths et al. and then a different citation.

• Citation [48] regards the paper on tree finches mentioned in the sentence, where the authors use the Griffiths et al. primers. We agree the format is confusing and the sentence now reads as follows: “The only report on PCR sexing of Darwin’s finches we are aware of uses primers described by Griffiths et al. [26] and includes 68 juvenile tree finches [48].” (Line 261-263)

---

## [Decision Letter · Decision Letter 1]

16 Feb 2021

Sex identification in embryos and adults of Darwin’s finches

PONE-D-20-22782R1

Dear Dr. Dobreva,

We’re pleased to inform you that your manuscript has been judged scientifically suitable for publication and will be formally accepted for publication once it meets all outstanding technical requirements.

Kind regards,

Anton Wutz

Academic Editor

PLOS ONE

Additional Editor Comments (optional):

Reviewers' comments:

Reviewer's Responses to Questions

**Comments to the Author**

1. If the authors have adequately addressed your comments raised in a previous round of review and you feel that this manuscript is now acceptable for publication, you may indicate that here to bypass the “Comments to the Author” section, enter your conflict of interest statement in the “Confidential to Editor” section, and submit your "Accept" recommendation.

Reviewer #2: All comments have been addressed

Reviewer #3: All comments have been addressed

2. Is the manuscript technically sound, and do the data support the conclusions?

Reviewer #2: Yes

Reviewer #3: Yes

3. Has the statistical analysis been performed appropriately and rigorously? 

Reviewer #2: Yes

Reviewer #3: N/A

4. Have the authors made all data underlying the findings in their manuscript fully available?

Reviewer #2: Yes

Reviewer #3: Yes

5. Is the manuscript presented in an intelligible fashion and written in standard English?

Reviewer #2: Yes

Reviewer #3: Yes

6. Review Comments to the Author

Reviewer #2: This is the second round of revision and I carefully read the new version of the paper by Dobreva and colleagues.

Dobreva and colleagues addressed the comments raised by the reviewers and at this stage I believe that the article is suitable to be published in PlosOne.

I have no further concerns regarding the paper.

With my best regards.

Reviewer #3: Some minor grammatical errors remain such as misplaced commas, but the authors have addressed all my concerns so I am happy with the state of the manuscript.

7. PLOS authors have the option to publish the peer review history of their article (what does this mean?). If published, this will include your full peer review and any attached files.

Reviewer #2: No

Reviewer #3: No

---

## [Editor Report · Acceptance letter]

22 Feb 2021

PONE-D-20-22782R1 

Sex identification in embryos and adults of Darwin’s finches 

Dear Dr. Dobreva:

I'm pleased to inform you that your manuscript has been deemed suitable for publication in PLOS ONE. Congratulations! Your manuscript is now with our production department. 

Kind regards, 

on behalf of

Dr. Anton Wutz 

Academic Editor

PLOS ONE